# Amplified Drought Alters Leaf Litter Metabolome, Slows Down Litter Decomposition, and Modifies Home Field (Dis)Advantage in Three Mediterranean Forests

**DOI:** 10.3390/plants11192582

**Published:** 2022-09-30

**Authors:** Elodie Quer, Susana Pereira, Thomas Michel, Mathieu Santonja, Thierry Gauquelin, Guillaume Simioni, Jean-Marc Ourcival, Richard Joffre, Jean-Marc Limousin, Adriane Aupic-Samain, Caroline Lecareux, Sylvie Dupouyet, Jean-Philippe Orts, Anne Bousquet-Mélou, Raphaël Gros, Marketa Sagova-Mareckova, Jan Kopecky, Catherine Fernandez, Virginie Baldy

**Affiliations:** 1Aix Marseille University, Avignon University, CNRS, IRD, IMBE, 13397 Marseille, France; 2CNRS, Nice Institute of Chemistry, UMR 7272, Parc Valrose, University of Côte d’Azur, 06108 Nice, France; 3INRAE, Ecologie des Forêts Méditerranéennes (UR629) Domaine Saint Paul, Site Agroparc, 84914 Avignon, France; 4CNRS, EPHE, IRD, CEFE, University Paul Valéry Montpellier, 34090 Montpellier, France; 5Department of Microbiology, Nutrition and Dietetics, Faculty of Agrobiology, Food and Natural Resources, Czech University of Life Sciences, Kamýcká 129, 165 00 Praha, Czech Republic; 6Crop Research Institute, Drnovska 507, 16106 Praha, Czech Republic

**Keywords:** experimental drought, Home Field Advantage (HFA), litter quality, Mediterranean forest, metabolomics

## Abstract

In Mediterranean ecosystems, the projected rainfall reduction of up to 30% may alter plant–soil interactions, particularly litter decomposition and Home Field Advantage (HFA). We set up a litter transplant experiment in the three main forests encountered in the northern part of the Medi-terranean Basin (dominated by either *Quercus ilex*, *Quercus pubescens*, or *Pinus halepensis*) equipped with a rain exclusion device, allowing an increase in drought either throughout the year or concentrated in spring and summer. Senescent leaves and needles were collected under two precipitation treatments (natural and amplified drought plots) at their “home” forest and were left to decompose in the forest of origin and in other forests under both drought conditions. MS-based metabolomic analysis of litter extracts combined with multivariate data analysis enabled us to detect modifications in the composition of litter specialized metabolites, following amplified drought treatment. Amplified drought altered litter quality and metabolomes, directly slowed down litter decomposition, and induced a loss of home field (dis)advantage. No indirect effect mediated by a change in litter quality on decomposition was observed. These results may suggest major alterations of plant–soil interactions in Mediterranean forests under amplified drought conditions.

## 1. Introduction

Terrestrial plants produce about 100 Gt of biomass annually at a global scale [1], 90% of which returns to the soil as leaf litter organic matter [2]. Litter is the basis of a complex food web which controls nutrient turnover, carbon (C) sequestration or mineralization, and the overall ecosystem functioning [3].

Litter quality [4,5] and environmental factors (i.e., climate, [6,7,8]) affect the decomposer community (including bacteria, fungi, and fauna), and in consequence, litter decomposition efficiency. The interaction between litter quality, environmental conditions, and soil decomposers can be explained by the Home Field Advantage (HFA) hypothesis: litter can be decomposed faster in its own habitat (at ‘home’) than away from it [9,10], due to the presence of specialized local decomposers [11]. Thus, local litter can be an important “selection force” on the activity and composition of the associated decomposer community [12]. An HFA hypothesis has been applied worldwide to explain litter decomposition in different types of habitats from boreal coniferous to tropical broadleaf forests [13,14]. In a synthesis of 35 reciprocal leaf litter transplantation experiments measuring decomposition among a variety of tree species and forest ecosystems (in Europe and America), It was demonstrated that, on average, the mass loss was 8% faster “at home” than in “away” habitats for a decomposition period up to 2 years [13]. However, in a litter transplant experiment between forest and meadow in New Zealand, it showed no evidence of HFA despite large phenotypic variations in leaf traits (i.e., morphological, physiological, and chemical traits) after 9 months of field decomposition [15]. These results emphasize the importance of environmental context and soil organisms in decomposition dynamics [15]. Nevertheless, and contrarily to soil organisms, the effects of climate on HFA are just beginning to be unveiled [16].

The functioning of Mediterranean terrestrial ecosystems is particularly constrained by water availability [17]. Current climate change scenarios predict an increase in temperature and a reduction in annual rainfalls in the Mediterranean region, particularly during summer [18]. This may aggravate abiotic conditions for soil organisms (i.e., microbial and faunal communities, [19,20,21]). During summer, soils could be drier more frequently and for a longer time. Aggravated drought conditions can directly affect litter decomposers, as it has been demonstrated that enzymatic activities and microbial respiration were negatively affected by water shortage [22,23,24] leading to a negative effect on the soil food web and then the decomposition process [25]. Moreover, aggravated drought conditions can indirectly affect decomposition through changes in litter quantity (e.g., reduction in leaf production; [24,26]) but also litter quality [19,27,28]. The intensification of summer drought may induce changes in plant metabolome to cope with these constraints [29,30,31]. These changes can be associated with a potential increase in the concentration of specialized metabolites such as phenolics and terpenes which show thermo-, oxido-, and photoprotective properties reducing (cell) oxidation processes under water stress conditions [32,33,34]. Both phenolics and terpenes are also known as allelopathic compounds that can affect either neighbor’s plants or soil microbial and faunal activity [35,36,37]. Moreover, such specialized metabolites are known to contain aromatic rings that are difficult to degrade and therefore reduce litter decomposability [38]. In the longer term, plant community composition may also be shifted towards communities of more drought tolerant species which may lead to more recalcitrant litter types (e.g., lignin-rich species, or sclerophyllous vegetation; [39]).

Thus, both the direct and indirect effects of water limitation could lead to a decrease in the litter decomposition rates of Mediterranean plant species [38,40], altering plant–soil interactions and potentially HFA [41]. Nevertheless, to the best of our knowledge, the effect of climate change on HFA has never been tested before in Mediterranean ecosystems.

This study aims to assess the potential effects of predicted drought intensification on litter decomposition and HFA effects in the Mediterranean region. For that purpose, we set up a two-year in situ crossed experiment in the three main Mediterranean forest types of southern France: (i) downy oak forest (*Quercus pubescens* Willd.), highly represented in the Mediterranean basin and which could become scarce in response to climate change [42]; (ii) holm oak forest (*Quercus ilex* L.), a key species concerning the structure of most forests in the western Mediterranean [43] and (iii) Aleppo pine forest (*Pinus halepensis* Mill.), a pioneer species with a current expansion mainly in the agricultural abandoned lands [43]. Each of these forest sites is equipped with a rain exclusion device simulating the increase in aridity conditions according to the climate change predictions for the Mediterranean region [18]. We studied the shift in the metabolism and elemental composition of litter from the three tree species to obtain a broad overview of litter quality changes under amplified drought. Our hypotheses were that (i) litter decomposition dynamics vary according to the type of Mediterranean forests studied, (ii) litter decomposition is negatively affected by amplified drought through a direct effect on decomposer activity and an indirect effect through a decrease in litter quality (e.g., increase in specialized metabolites and decrease in nutrient contents), (iii) there is an HFA for litter decomposition in Mediterranean forests that is altered by amplified drought.

## 2. Results

### 2.1. Initial Litter Quality

The initial litter quality of the litter coming from ND plots varied significantly among the three litter species (Table 1).

*Q. ilex* litter exhibited higher concentrations of N, P, K, and cellulose but a lower WSC concentration and SLA compared to the two other species (ANOVAs, *p* < 0.001). *Q. pubescens* litter showed higher concentrations of Ca and Mg but lower C concentrations compared to the litter of the two other species (ANOVAs, *p* < 0.001). In addition, *Q. pubescens* litter showed the highest SLA and WHC values (ANOVAs, *p* < 0.001). *P. halepensis* litter showed higher C concentrations and lower WHC, N, P, and Ca concentrations compared to the litter of the two oak species (ANOVAs, *p* < 0.001).

Initial litter traits also varied between leaves and needles collected in AD or ND plots (Table 1). Amplified drought induced a reduction in Ca concentration for the three litter species (ANOVAs, *p* < 0.001), a reduction in N concentration only for *Q. ilex* litter, and a reduction in Na concentration only for *P. halepensis* litter (ANOVAs, *p* < 0.01). With amplified drought, Mg and Na concentrations increased for *Q. ilex* and *Q. pubescens* litters, while K and phenolic concentrations increased only for *Q. ilex* litter (ANOVAs, *p* < 0.001). P concentration increased only for *P. halepensis* litter (ANOVAs, *p* < 0.01). WSC decreased in AD plots for *Q. pubescens* and *P. halepensis* litters (ANOVAs, *p* < 0.01), while WHC and SLA decreased for *Q. pubescens* and *Q. ilex* litters (ANOVAs, *p* < 0.05). No change in C, cellulose, hemicellulose, and lignin concentrations were observed (ANOVAs, *p* > 0.05).

### 2.2. Initial Metabolomic Litter Signature

MS-based metabolomic analysis of litter extracts combined with multivariate data analysis (PCAs and PERMANOVAs) enabled us to detect modifications in the composition of litter-specialized metabolites according to litter types (i.e., senescent leaves and needles collected in AD or ND plots). A total of 850, 1105, and 900 molecular ions were detected in the *Q. ilex*, *Q. pubescens*, and *P. halepensis* litter extracts, respectively. As observed on the PCA, the overall metabolites composition of these litter extracts varied according to the litter type (Figure 1).

Multivariate permutation analysis confirmed that the litter metabolomes significantly differed if the leaves came from AD or ND plots for *Q. ilex* and *Q. pubescens* (Table 2, Figure 1, PERMANOVAs, *p* < 0.05).

The metabolomic profile of *P. halepensis* needles was not significantly different between needles from AD and ND plots (Table 2, Figure 1, PERMANOVA, *p* > 0.05). The fifteen highest Variable Important in Projection (VIP) were tentatively annotated for *Q. ilex* and *Q. pubescens* leaves (Table 3 and Table 4) while specialized metabolites from *P. halepensis* needles were not annotated (PERMANOVA, *p* > 0.05).

A combination of accurate mass, isotope ratio, MS/MS fragmentation, and public libraries was used to annotate the metabolites detected by an untargeted metabolomic approach (see the Appendix A for further information on metabolites annotation, Appendix A). A comparison of VIP retention time reveals that biomarkers are distributed over the entire chromatogram, suggesting a huge difference in terms of structure and polarity. Detected metabolites in *Quercus* litter submitted to AD generally belonged to phenolics. For instance, *Q. ilex* litter putatively contained flavonols (e.g., Quercetin-3-O-(6″-O-galloyl)-b-D-glucopyranoside, Isorhamnetin 3-(6″-galloylglucoside), glycoside coumarin (e.g., aesculin), and cinnamic acid derivatives (e.g., chlorogenic acid). Several phenolics were also detected in *Q. pubescens* litter submitted to AD such as tannins (e.g., Proanthocyanidin, Galloylquinic acid isomer) and flavanols (e.g., (epi)catechin). *Q. pubescens* litter submitted to AD was also characterized by other kinds of metabolites such as triterpenoids (e.g., Arjungenin, Oleane triterpene). It is worthwhile to note that one compound (M387T49) was found in both leaf extracts of *Q. ilex* and *Q. pubescens* collected in ND plots. According to its formulae and its typical MS/MS fragmentation, it was identified as a disaccharide. Finally, even if most of the metabolites annotated were phenolics and terpenoids, several VIPs occurring in litter extracts remain unidentified as no MS2 spectra could be recorded for them or they have never been described in the literature (Appendix A).

### 2.3. Leaf Litter Decomposition

Litter decomposition varied according to the decomposition time, precipitation treatments, litter species, and forests (Table 5, Figure 2), while no effect of litter type was observed.

Due to the absence of the litter type effect (Table 2), the two litter types (i.e., senescent leaves and needles collected in ND and AD plots) were pooled. The effects of precipitation treatments, litter species, and forests on litter mass loss were dependent on decomposition time (Table 5). Litter decomposition was significantly 4.0% lower under amplified compared to natural drought conditions after two years of decomposition (significant precipitation treatments × time interaction, Table 2; Figure 2a). *P. halepensis* showed higher litter mass loss than the two other species after one year, while *Q. ilex* and *Q. pubescens* showed higher litter mass loss than *P. halepensis* after two years of decomposition (significant litter species × time interaction, Table 5; Figure 2b). After two years of decomposition, the mass loss was significantly higher in Puechabon (*Q. ilex* forest, 44.8%) than in the other two forests (40.9% and 39.2% in Fontblanche and in O_3_HP *P. halepensis* and *Q. pubescens* forests, respectively) (significant forests × time interaction, Table 5; Figure 2c).

### 2.4. Home Field Advantage (HFA)

In ND plots, we observed an HFA for *P. halepensis* only after 2 years of decomposition, with 4.4% higher litter mass remaining “away” than “at home” (Table 6).

On the opposite, we observed an HFD in ND plots for *Q. pubescens* with 1.6% and 9.4% higher litter mass remaining “at home” than “away” after one and two years of decomposition, respectively. In AD plots, *P. halepensis* loses its HFA and *Q. pubescens* loses its HFD after two years of decomposition (Table 6).

## 3. Discussion

This work is the first to study the HFA of the dominant tree species in the north of the Mediterranean basin in a climate change context. Our results showed that amplified drought: (i) impacts leaf litter quality such that litter decomposition ability was negatively affected, (ii) has a direct effect on litter decomposition by reducing litter mass loss but, in contrast, has no indirect effect mediated through a shift in litter quality, and (iii) alters HFA and HFD effects. However, the intensity of the effects is highly dependent on the litter species identity, site environmental conditions, and the duration of exposure to amplified drought.

### 3.1. Amplified Drought Alters Initial Litter Metabolomes and Initial Litter Quality

The initial litter traits measured for *P. halepensis*, *Q. pubescens*, and *Q. ilex* were consistent with those observed in previous studies [36,40,55]. According to our hypothesis, the litter quality of the three species was altered under amplified drought, and this alteration led to a decrease in litter quality, at such a point that initial metabolomes differed according to litter types. However, these effects varied greatly according to the litter species and litter traits considered.

The AD treatment decreased leaf and needle Ca (for all species) and N (for *Q. ilex*) concentrations. This may be explained by (1) nutrient resorption from green leaves or needles [56] and reallocation to other plant organs [57], (2) the reduction in transpiration fluxes and turgor, which may decrease element absorption by the roots and consequent transport from the roots to the leaves or needles [58], and (3) the inhibition of enzymes involved in nutrient metabolism [57,59]. On the other hand, we observed an increase in K (for *Q. ilex*), Na, and Mg (for *Q. ilex* and *Q. pubescens*) concentrations under amplified drought. The same results were obtained by [57], who observed that the increase in leaf Mg concentrations was accompanied by an increase in the soil capacity to mobilize this element after 6 years of amplified drought exposition, suggesting a faster Mg cycling between plants and soil under drier conditions. K and Ca have an important role in controlling plant cell turgor pressure and thus may help to reduce water losses under drier conditions [48]. This may suggest a water stress avoidance mechanism in these *Quercus* species under increasing water stress. Increases in Na concentration with drier conditions have been already observed in arid and semi-arid ecosystems [59,60]. Furthermore, the increase in Na concentration may be linked with the decrease in Ca concentration. Indeed, [60] reported that adjustments of the osmotic pressure of *Populus euphratica* leaves were achieved by a Na accumulation of and a compensatory decrease in Ca and soluble carbohydrates to cope with the stress caused by the salinity in an arid ecosystem.

The phenolic concentration increased in *Q. ilex* leaves, suggesting a different C allocation under amplified drought. According to the growth-differentiation balance hypothesis (GDBH), there is a physiological trade-off between growth and plant defense [61]. This hypothesis predicts that any environmental factor that constrains growth more than photosynthesis can differentiate the allocation of internal resources toward the production of specialized metabolites. Linked to the decrease in N content, we confirmed that amplified drought alters this trade-off for *Q. ilex*. Moreover, drought biomarkers in *Q. ilex* and *Q. pubescens* leaves highlighted by metabolomic analysis were mostly polyphenols, particularly flavonoids and their derivatives (e.g., (epi)catechin, proanthocyanidin, isorhamnetin 3-(6″-galloylglucoside)). These results were consistent with those of [48], showing that an amplified drought treatment for 10 years induced a higher foliar concentration of flavonoids for *Q. ilex*. Flavonoids have antioxidant properties due to their capacity to scavenge free radicals such as Reactive Oxygen Species (ROS) [62]. Thus, we can suggest that flavonoids provided antioxidant protection against stress induced by amplified drought in oak leaves. Moreover, we identified triterpenoid derivatives (e.g., arjungenin glycoside isomer 1 and oleane triterpene) in *Q. pubescens* leaves. Specialized triterpenoid metabolites are usually involved in plant defense and development. For instance, triterpenoid (i.e., lupeol) accumulated in the cuticular wax surface was suggested to have a physiological role in protection against dehydration [63]. Triterpenoids may also play a role in allelopathic interactions with neighboring plants. It was observed that triterpenoids (i.e., pentacyclic triterpenoids) contained in *Alstonia scholaris* litter significantly inhibited the growth of *Bidens pilosa*, a weed growing abundantly in *Alstonia scholaris* forests [64].

The two oak species showed a decrease in SLA in AD plots. The SLA reduction with rainfall reduction has been demonstrated in previous studies [19,65]. The reduction in SLA may suggest (1) a decrease in water loss through transpiration and leaf surface temperature [66,67] and (2) a change in C allocation [65]. It was demonstrated that savanna species allocated more biomass to roots and lower SLA than well-sun-exposed forest species [65]. Specialized metabolite composition was also modified in *Q. ilex* and *Q. pubescens* senescent leaves which may also suggest an investment in defense compounds over tissue growth. A study showed that an increase in specialized metabolite production in *Q. pubescens* was accompanied by a decrease in plant growth after 2 years of field stress conditions [68]. These results also showed that the time of exposure and the intensity of the water constraints, which depend on the climate of the study site (Table 7), have an important impact on the chemical and physical litter traits of these three Mediterranean species. *Q. pubescens* and *P. halepensis* trees were exposed to amplified drought for a shorter time (4 years and 7 years, respectively) than those in *Q. ilex* forest (13 years). Contrary to the two *Quercus* species, amplified drought treatment for 7 years induced no significant change in the initial metabolome of *P. halepensis* litter. Pine litter is rich in structurally recalcitrant and therefore very stable compounds such as lignin [69]. These properties may explain a certain stability of the compounds contained in these needles before and during decomposition independently of the amount of precipitation.

### 3.2. Decomposition Dynamics in the Three Mediterranean Forests

On average, *P. halepensis*, *Q. ilex*, and *Q. pubescens* lost 42%, 50%, and 52% of initial litter mass after 2 years of field exposure, respectively. These rates of litter decomposition are similar to those reported in other studies from Mediterranean forested ecosystems [36,40,70]. It was reported that around 50% of the mass was lost for *Q. pubescens* after 25 months of field exposure [71]. Additionally, a mass loss of around 40% was shown for *P. halepensis* after 700 days [72], and about a 53% mass loss was observed for *Q. ilex* after 2 years [55].

In the present study, the differences in litter decomposition rates between the three tree species varied significantly between the first and the second year of decomposition, suggesting that distinct litter traits control the decomposition process in the early compared to later decomposition stages. After one year of decomposition, the litter of *P. halepensis* decomposed better than the litter of *Q. pubescens* and *Q. ilex* (62% of remaining litter mass versus 67 and 65%, respectively), which may be due to a higher level of water-soluble compounds (WSC) in its senescent needles. It has been shown that there is a positive and significant correlation between WSC and cumulative loss during the decomposition process [73]. This could be explained by the fact that WSCs could act as a readily decomposable substrate for decomposers [74]. After two years of decomposition, *Q. pubescens* and *Q. ilex* litters were better decomposed than *P. halepensis* (49% and 50% of remaining litter mass, respectively, versus 58%). The higher nutrient concentrations (N, P, Ca, and Mg) in the two oak species compared to *P. halepensis* may improve decomposer growth and activity, leading to a higher litter decomposition after two years. Mg and Ca have been considered as key drivers of litter C loss across biomes [4]. In fact, litter Ca has been reported as an essential co-factor of the ligninolytic enzyme activity and it is also known to favor fungal growth [75] as well as faunal communities [76]. The two oak species exhibit also a higher water holding capacity (WHC) than *P. halepensis*, a physical trait directly affecting litter humidity that is highly important to maintaining decomposer activity [5,40,77].

In our study, litter decomposition was only 4% lower in AD plots compared to ND plots after two years of decomposition. This result is nevertheless congruent with previous studies conducted in Mediterranean ecosystems that also found a weak reduction in decomposition rates in the treatments with reduced water availability [21,40,78]. Because humidity is a key environmental factor for microbial [20,72] and faunal activities [79], we expected a stronger negative effect of reduced precipitation on the decomposition process. The weak effect observed may be due to the unique conditions of the Mediterranean climate, with great seasonal shifts in both temperature and soil water availability and the rather extreme environmental conditions during summer [80], which could represent an important selection pressure for decomposer communities, resulting in higher drought tolerance [81]. Such enhanced drought tolerance of decomposer communities presumably explained the weak negative effects on the decomposition observed in the present study compared to other studies performed in temperate [17,82] or tropical forests [17] that presented stronger effects of reduced precipitation.

Surprisingly, despite the modifications of litter traits and metabolomes (for *Q. ilex* and *Q. pubescens*) between trees from ND and AD plots, we did not observe an indirect effect of amplified drought on the litter decomposition process mediated by this shift in litter quality. This key finding indicates that climate change would alter directly the decomposition process by affecting the decomposer activity rather than modifying the quality of the litter that they decompose in the three studied Mediterranean forests.

### 3.3. Home Field Advantage in the Three Mediterranean Study Forests

According to our third hypothesis, *P. halepensis* litter showed a significant HFA after 2 years of field decomposition. This result is consistent with the hypothesis that HFA increases when litter quality decreases due to the presence of specialized decomposers [13]. In low-quality litter such as pine litter that contains complex recalcitrant compounds and toxic secondary metabolites [36], HFA is generally more important because only very specific decomposer communities and enzymes can degrade it rapidly [13,83]. Fungal activity in pine species litter decomposition is widely documented [36,84,85] and fungal species produce a wide range of extracellular enzymes [86], which may be an advantage in the presence of high chemical diversity. However, enzymatic performance increases with increasing temperature until a limiting threshold [87,88] and decreases with lower temperatures [88]. Thus, the mild winters in the *P. halepensis* forest (Meso-Mediterranean bioclimatic zone) may favor enzymatic activity and decomposer activity in this forest, in comparison to the low temperature during winter in *Q. pubescens* forest (Supra-Mediterranean bioclimatic zone). Our results on litter decomposition in the three ND plot forests confirm that the *Q. pubescens* forest is the least favorable for litter decomposition than the two other forests.

The litter of *Q. Ilex* showed the lowest C:N ratio compared to the two other species, which can justify the fact that there is no HFA observed since they may be consumed by generalist decomposer communities present in the three forests. Thus, *Q. ilex* decomposition occurs as well “at home” as “away”. Regarding the decomposition of *Q. pubescens* litter, the results obtained go against our hypothesis. We observed a “Home Field Disadvantage” (HFD) meaning that this litter decomposes better “away” than “at home”. This may be due to the fact that the input of *Q. pubescens* litter with high SLA and WHC in drier sites (such as *P. halepensis* forest) may lead to litter decomposition more efficient away than at home [5,89].

Finally, according to our last hypothesis, HFA and HFD were altered under amplified drought conditions. For instance, the HFA disappeared under amplified drought conditions for *P. halepensis* species. This may corroborate a study that showed that drought had no impact on microbial specialization in different litter types after one year of litter incubation [90]. In addition, the analysis of 125 reciprocal litter transplants from 35 studies found that, although climate is one of the major drivers of litter mass loss, it does not emerge as an important determinant of HFA [16]. Nevertheless, the intensification of water stress conditions in an already dry site (*P. halepensis*) may negatively impact soil decomposers and contribute to lower pine needle decomposition. On the other hand, the improvement of microclimatic conditions “away” may favor its decomposition in foreign habitats.

## 4. Material and Methods

### 4.1. Study Sites

The litter decomposition experiment was performed in three AnaEE experimental sites: Puechabon holm oak forest (*Quercus ilex*), the Oak Observatory at the “Observatoire de Haute Provence” (O_3_HP) downy oak forest (*Quercus pubescens*), and Fontblanche Aleppo pine forest (*Pinus halepensis*), located in the South of France (Table 7, Appendix A).

To simulate the extension and intensification of the summer drought period forecasted by the climatic models in the three Mediterranean forests, a rain exclusion plot (submitted to Amplified Drought [AD] corresponding to the reduction in annual rainfall events in 30% projected by the climatic models, RCP 8.5 scenario, [18,91]) was settled next to a control plot (submitted to the Natural Drought [ND] conditions).

In the downy oak forest, the rain exclusion was started in 2012 by the implementation of a 15 m × 20 m rainout shelter above the canopy which dynamically excluded precipitations by deploying automated shutters during rainfall events of the vegetation period (i.e., from spring to autumn). In the holm oak and Aleppo pine forests, the rain exclusion is performed by using fixed PVC gutters (in an area of 25 m × 25 m) installed below the forest canopy, excluding about 30% of each rainfall event throughout the year. Control plots have the same system, but the PVC gutters were fixed upside down to not exclude rainfall. The corresponding rain exclusion plots were settled in 2003 and 2008 in the holm oak and Aleppo pine forests, respectively.

During the first year of the decomposition experiment (2015), the mean annual temperatures were 14.01, 13.7, and 12.1 °C, while in the second year (2016), they were 14.2, 13.9, and 12.4 °C in the Aleppo pine, holm oak, and downy oak forests, respectively. The mean annual precipitation during the first year was 631.8, 794.3, and 619.8 mm and during the second year, 615.6, 1184.2, and 804.4 mm for the Aleppo pine, holm oak, and downy oak forests, respectively (Appendix A). The devices allowed the exclusion of 33% and 42% for the first and second years in the downy oak forest and approximately 30% in the two other forests.

### 4.2. Experimental Setup and Litter Bag Processing

Senescent leaves were collected in ND and AD plots of the three forests (Figure 3). For that, litter traps were used during the abscission period that occurred from April to May 2014 for the *Q. ilex* leaves, from June to September 2014 for the *P. halepensis* needles, and from October to November 2014 for the *Q. pubescens* leaves. The leaves and needles were air dried and stored at room temperature in paper boxes until the beginning of the experiment.

Litter decomposition was assessed by using the litter bag method [92]. Briefly, 4 mm mesh litter bags (20 × 20 cm) containing 10 g (air-dried) of the senescent material were used to perform the experiment. Firstly, in order to test the direct and indirect effects of AD on decomposition, litter bags composed of leaves or needles coming from trees on AD and ND were reciprocally placed on the AD and ND plots of the three forests. Secondly, in order to test the HFA, litter transplants were performed between each site for the three species considered, i.e., a litter bag containing the litter of each species placed on each forest. Finally, to test the effect of AD on HFA, both types of litter transplants were combined (Figure 3).

Thus, the experiment consisted of 36 modalities corresponding to the three forests (holm oak, downy oak, and Aleppo Pine) × three litter species (*Q. pubescens, Q. ilex,* and *P. halepensis*) × two litter types (senescent leaves and needles from AD and ND plots) × two precipitation treatments (AD and ND). In total, 504 litter bags (36 modalities × 2 sampling dates × 7 replicates) were analyzed. Litter bags were placed perpendicularly to the gutters system in the Aleppo pine and holm oak forests and under the rain exclusion device in the downy oak forest by using a transect in blocks (7 columns × 12 lines), equidistant from each other (1 m distance between the seven columns and 0.6 m between the 12 lines). The transects were E-W oriented. They were placed on the ground floor after the removal of the litter layer and fixed to the soil with galvanized nails to prevent movement by animals or wind. The litter layer was then replaced over the litter bags.

The litter mass loss was measured after one and two years of decomposition. At each sampling date, 84 litter bags (i.e., half of the litter bags at each forest) were retrieved from each forest. The litter bags were placed in plastic bags to prevent the loss of biological material.

In the laboratory, the litter contained in the litter bags was cleaned from soil and other plant detritus and stored at −18 °C. All the samples were freeze-dried (Lyovac GT2), weighed, and ground to powder (MM400, Restch Inc. Newtown, PA, USA) for chemical analysis. At the beginning of the experiment, 30 samples (3 litter species × 2 litter types × 5 replicates) were used to determine the initial dry weight (g).

### 4.3. Initial Litter Quality

The initial litter traits of the three litter species (*Q. pubescens*, *Q. ilex*, and *P. halepensis*) were determined using five samples of each litter type (i.e., senescent leaves and needles collected from trees in AD or ND plots; Figure 3).

The total C and nitrogen (N) concentrations were determined by thermal combustion on a Flash EA 1112 series C/N elemental analyzer (Thermo Scientific^®^, Waltham, MA, USA). Total Phosphorus (P), calcium (Ca), Potassium (K), Magnesium (Mg), and sodium (Na) were extracted from 80 mg of ground litter sample with 8 mL of nitric acid and 2 mL of H_2_O_2_ at 175 °C for 40 min using a microwave digestion system (Ethos One, Milestone SRL, Sorisole, Italy). After the mineralization step, every sample was adjusted to 50 mL with demineralized water. The P concentration was measured colorimetrically by using the molybdenum blue method [93]. In total, 100 µL of sample, 100 µL of NaOH, 50 µL of mixed reagent (emetic tartar and ammonium molybdate solution), and 50 µL of ascorbic acid were mixed directly in a 96-well microplate. After 45 min at 40 °C, the reaction was completed, and the P concentration was measured at 720 nm using a microplate reader (Victor, Perkin Elmer, Waltham, MA, USA). The Ca, K, Mg, and Na concentrations were determined using an atom absorption spectrometer (AAS, iCE 3000 series, Thermo Scientific^®^, Waltham, MA, USA).

The lignin, cellulose, hemicellulose, and water-soluble compound (WSC) concentrations were determined according to the van Soest extraction protocol [94] using a fiber analyzer (Fibersac 24; Ankom^®^, Macedon, NJ, USA).

Total phenolic concentrations were measured by colorimetry according to the method of using gallic acid as standard [95]. Aqueous extracts were created by dissolving 0.25 g of litter powder in 20 mL of 70% aqueous methanol solution before being shaken for 1 h and filtered (0.45 μm). Briefly, 25 mL of the extracts obtained were then mixed with 0.25 mL of Folin–Ciocalteu reagent [96], 0.5 mL of saturated aqueous Na_2_CO_3_ (to stabilize the colorimetric reaction), and 4 mL of distilled water. After 60 min, the reaction was completed and the concentration of phenolics was measured at 765 nm using a UV/Vis spectrophotometer (Thermo Scientific^®^, Waltham, MA, USA).

The specific leaf area (SLA) was determined by using the Image J software (https://imagej.nih.gov/ij/, Version 1.46r, Wayne Rasband, Bethesda MA, USA, accessed on 1 January 2017). The SLA was calculated as the ratio between the leaf area and the leaf dry weight. To determine the water holding capacity (WHC), intact leaf litter samples were soaked in distilled water for 24 h, drained, and weighed (wet weight). The dry weight was measured after drying the samples for 48 h at 60 °C. The WHC (in percentage) was calculated according to the formula: (wet weight/dry weight) × 100 [40].

### 4.4. Initial Metabolomic Litter Signature

The extraction of the metabolites was performed on a total of 30 samples of litter (3 litter species × 2 litter types × 5 replicates). A total of 0.2 g of litter powder of each sample was extracted in 2 mL of a MeOH/water 1:1 (*v*/*v*) mixture, followed by an ultrasound bath for 3 min at room temperature. The extract obtained was then filtered for analysis (PTFE, 0.22 μm, Restek^®^, Bellefonte, PA, USA).

The metabolomic litter signature was acquired on a UHPLC instrument (Dionex Ultimate 3000, Thermo Scientific^®^,Waltham, MA, USA) equipped with an RS pump, an automatic sampler, and a thermostatically controlled column oven, and coupled to a Photodiode Array Detector (PDA), as well as an accurate quadrupole Time of Flight (qToF) mass spectrometer equipped with an Electrospray Ionisation ESI source (Impact II, Bruker Daltonics^®^, Billerica, MA, USA). The separation of the metabolites by UHPLC was carried out using a Luna^®^ Omega Polar C18 column (2.1 × 100 mm, 1.6 μm, Phenomenex^®^,Torrance, CA, USA). Gradient elution was performed with ultrapure water (Cogolin reagent FMP, Fluka) (A) and acetonitrile (Cogolin reagent FMP), both acidified with 0.1% formic acid (B) as follows: from 5% B during 2 min, then 20% B for 8 min following by a linear gradient up to 100% for 4 min, which was maintained for 3 min in isocratic mode. The analysis was followed by a return to the initial conditions for 3 min for a total analysis time of 20 min. The flow rate was set up at 0.5 mL/min at 45 °C. The injection volume was 0.2 µL for the *Q. ilex* and *P. halepensis* samples and 1 µL for the *Q. pubescens* samples. Chromatographic elution gradients were adjusted to improve peak resolution by using a mixture of all pooled samples. The litter extracts were injected in three separate runs according to the litter species and randomly according to the litter types. The pooled samples were injected to account for time-dependent MS drift, responsible for the variance in the MS results obtained. MS detection was performed in both negative and positive ionization modes. The MS parameters were defined as follows: nebulizer gas, N_2_ at 3.5 bar, N_2_ dry gas at 12 L min^−1^, capillary temperature at 200 °C, and voltage at 3000 V in negative mode and at −2500 V in positive mode. Analyses were recorded at an acquisition frequency of 4 Hz, and the mass range was set from *m*/*z* 50 to 1200 uma. The mass spectrometer was calibrated systematically with formate/acetate solution composed of 25 mL of ultrapure water, 25 mL of isopropanol (Sigma Aldrich, Darmstadt, Germany), 0.5 mL of NaOH, 25 μL of formic acid, and 75 μL of acetic. The acquired MS-based metabolomic data were calibrated automatically by internal calibration with a formate/acetate solution before exporting the data in netCDF files (centroid mode) using Bruker Compass DataAnalysis 4.3. All the converted analyses were then processed by the XCMS software [97] under R software version 3.5.1 [98], following different steps to generate the final data matrix: (1) Peak picking (peak width = c (2,20) Ppm = 2) without threshold prefilter [99]; (2) correction of the retention time (method = obiwarp), (3) grouping (bw = 10, minfrac = 0,3, minsamp = 1), (4) Fillpeaks, and then (5) report and generate the data matrix transferred to Excel. To remove technical variability, the matrix was then filtered according to blanks and pooled in three steps using in-house R scripts: (1) filtering the matrix using the Signal/Noise (S/N) ratio in order to remove the peaks observed in the blanks (S/N set at 10), (2) filtering the matrix using the peaks coefficient of variation in order to remove the peaks with variable intensities in the pooled samples (threshold at 0.3), (3) filtering the matrix according to autocorrelation between the peaks. MS/MS experiments were carried out with variable collision energies at 20 eV and 40 eV.

The metabolites were annotated using a combination of accurate mass, isotope ratio, MS/MS fragmentation, and public libraries. The putative identification of the compounds reported differentiating the metabolomes of *Quercus* senescent leaves collected in AD or ND plots was carried out by using a dereplication strategy [45,100]. Briefly, the DataAnalysis software was used to generate possible chemical formulae for the ions in the MS spectra. For reducing the possible candidates, C, H, and O were selected for elemental composition calculations, and only low values of mass accuracy (Δ ppm) and the isotopic pattern quality (mSigma value) were considered. Specific ions [M + HCOO]^−^, [M−2H]^2−^ and [M + H]^+^, [M + Na]^+^, respectively, observed in negative and positive modes, supported the confirmation of chemical formulae for the pseudomolecular ion [M − H]^−^. The possible molecular formulae for each selected peak were then searched in public databases, i.e., KnapSAcK, Dictionary of Natural Products, HMBD, and PubChem [101] to report known natural products and/or eliminate non-referenced formulae. Additionally, retention time (RT), HRMS/MS data, as well as bibliographic and chemotaxonomic information were employed for identification. The Bruker Compass DataAnalysis software (version 4.3, Bruker Daltonics^®^, Billerica, MA, USA) was also used to analyze and identify the compounds using MS and MS/MS spectra.

### 4.5. Data Analysis

All the statistical analyses were performed with the R software version 3.5.1 [98], with significance levels indicated as * for *p* < 0.05, ** for *p* < 0.01, and *** for *p* < 0.001.

Two-way analyses of variance (ANOVA) followed by Tukey tests were used to test the effects of the litter species (*Q. ilex*, *Q. pubescens*, or *P. halepensis*,) and litter types (i.e., senescent leaves and needles collected in AD or ND plots) and their interactions on initial litter traits.

Because the negative ionization mode presented a better sensitivity, data acquired in this mode were used to evaluate differences in metabolomes according to the litter species (*P. halepensis*, *Q. ilex*, or *Q. pubescens*) and litter type (i.e., senescent leaves and needles collected from trees in AD or ND plots). These differences were tested by using PERMANOVAs (function “adonis” in package “vegan”, [102]). The three associated Principal Component Analysis (PCAs, package “factoextra”, and function “fviz_pca_biplot” [103], on log-transformed and auto-scaled data) was performed separately for each litter species on untargeted metabolome datasets taking into account all metabolites (or detected features).

We used a general linear model approach to test for the effects of litter species (*P. halepensis*, *Q. ilex*, or *Q. pubescens*), litter types (i.e., senescent leaves and needles collected from trees in AD and ND plots), precipitation treatments (AD or ND), forest (*P. halepensis*, *Q. ilex* or *Q. pubescens* forests), and time of decomposition (1 or 2 years) on the litter mass remaining. The full model was simplified to determine the most parsimonious model by using an established model selection procedure with both forward and backward selection algorithms, which ranks all candidate models (all possible combinations of initial explanatory variables included in the full model) based on the lowest AICs (function “stepAIC” and “AIC” in package “MASS”, [104]). We present the r^2^ and AIC values for both the full model (with all initial explanatory variables) and the most parsimonious model.

The home field advantage (HFA) was calculated according to the method of [13] based on the following equations:
HDDi = (DiI − DjI) + (DiI − DkI)(1)
ADDi = (Dij − DjJ) + (DiK − DkK)(2)
H = (HDDi + HDDj + HDDk)/(N − 1)(3)
ADHi = HDDi − ADDi − H(4)
where ADHi (Additional Decomposition at Home) is the additional decomposition at home for the considered species: i, j, and k are litters derived from plant species; I, J, and K are the forests dominated by species i, j, and k. D is a measure of decomposition (i.e., remaining leaf dry mass), HDD and ADD represent the difference of decomposition ”at home” and ”away”; H is the total HFA for all species combined, and N is the number of plant species. If ADHi > 0, there is an HFA; if ADHi < 0, there is a Home Field Disadvantage (the decomposition of the litter “at home” is slower than ”away”); if ADHi = 0, there is no effect. To calculate the ADHi, twelve random permutations were performed for each species according to the treatment. Student tests were used to determine whether the ADH (1) differed from 0.

## 5. Conclusions

Amplified drought modifies litter quality and its metabolome, but this effect depends on the litter species considered and the time of exposure to drier conditions. For *Q. ilex* forest, submitted to amplified drought for 13 years, we observed that the increase in specialized metabolites (phenolic compounds) and changes in metabolic composition coincide with a reduction in litter morpho-anatomical and chemical traits (SLA, WHC, WSC, N, Mg, Ca). *P. halepensis* litter chemistry was so specific and recalcitrant that 7 years of amplified drought was not enough to significantly affect litter quality and metabolome. However, after only 4 years of amplified drought, *Q. pubescens* litter metabolome was altered, but the balance between primary and secondary metabolism was not clearly affected. Litter decomposition efficiency seems to be more impacted by the site of decomposition than litter quality. There is an important soil–plant affinity in *P. halepensis* and *Q. pubescens* forests manifested by significant HFA and HFD. The loss of these effects under rain exclusion suggests that local decomposers would reduce their ability to degrade *P. halepensis* and *Q. pubescens*.

## Figures and Tables

**Figure 1 plants-11-02582-f001:**
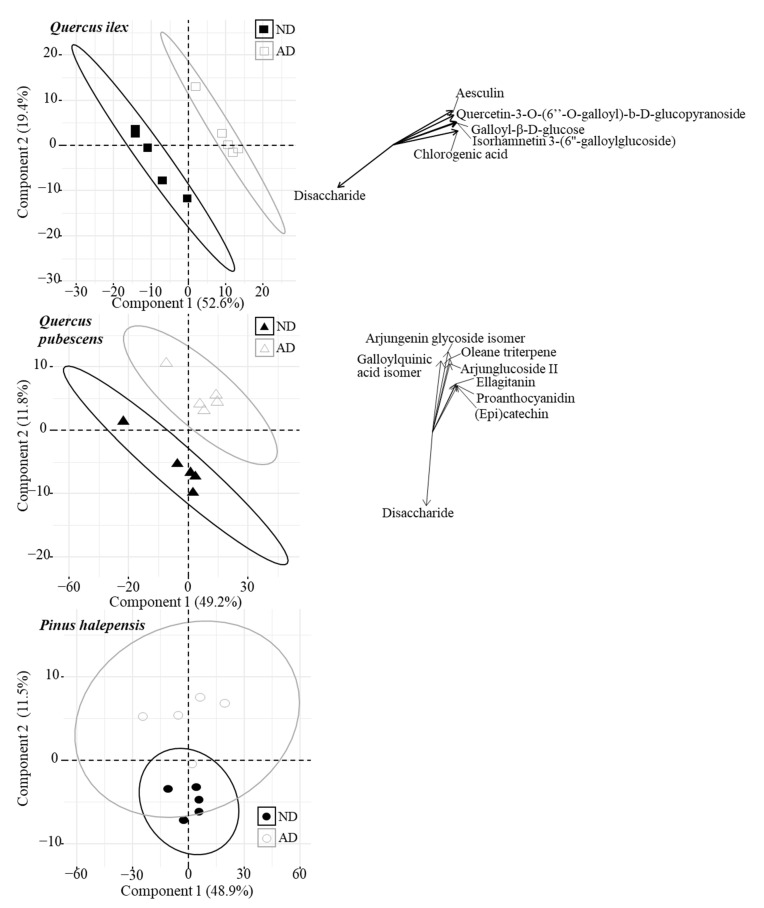
PCA of the initial metabolome of the litter of the three tree species (*Quercus ilex*, *Quercus pubescens*, and *Pinus halepensis*) in Natural Drought (ND) and Amplified Drought (AD) plots of the respective “home” forests.

**Figure 2 plants-11-02582-f002:**
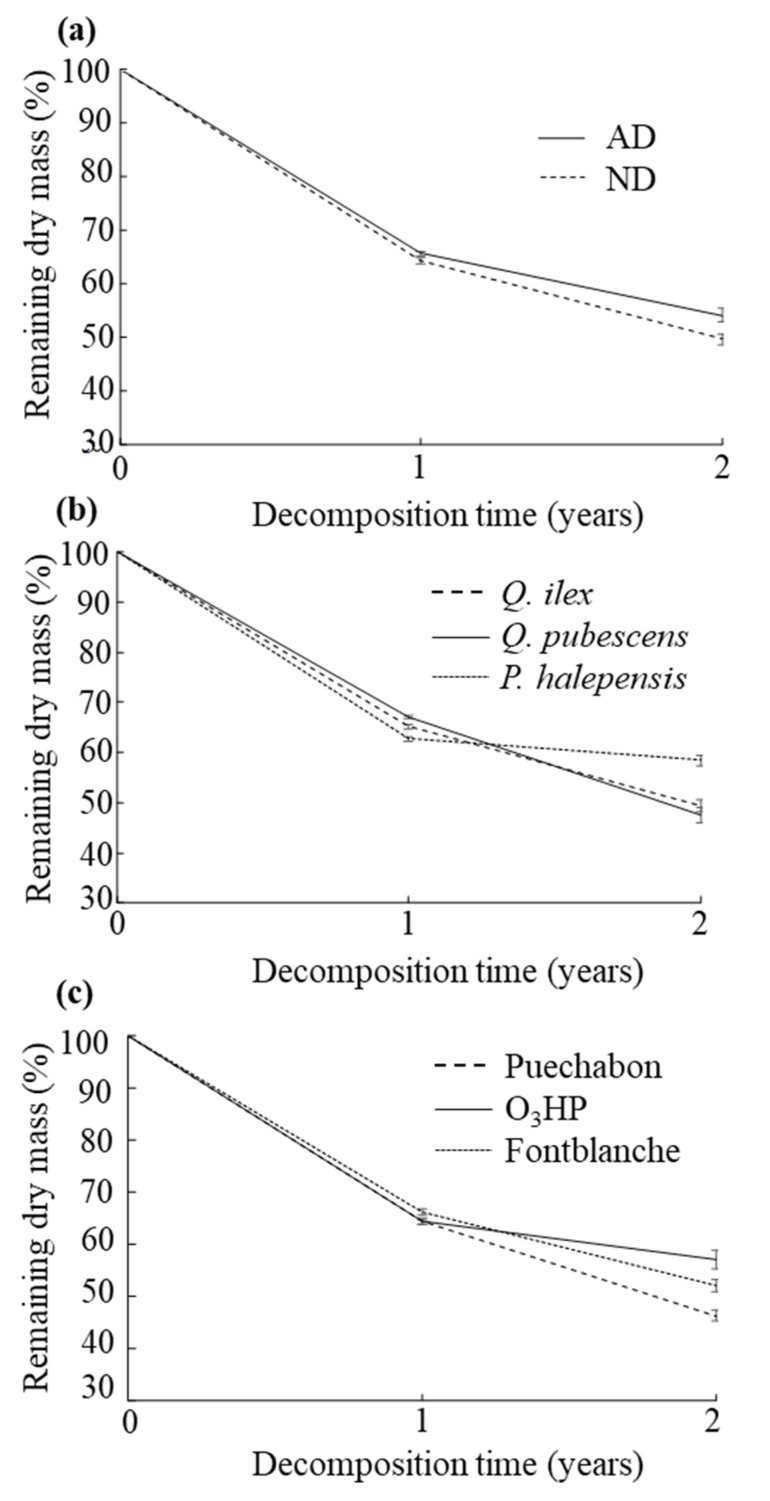
Dynamics of the remaining litter dry mass (in % of initial mass) during the two years of field decomposition, according to (**a**) precipitation treatments; (**b**) litter species; and (**c**) forests.

**Figure 3 plants-11-02582-f003:**
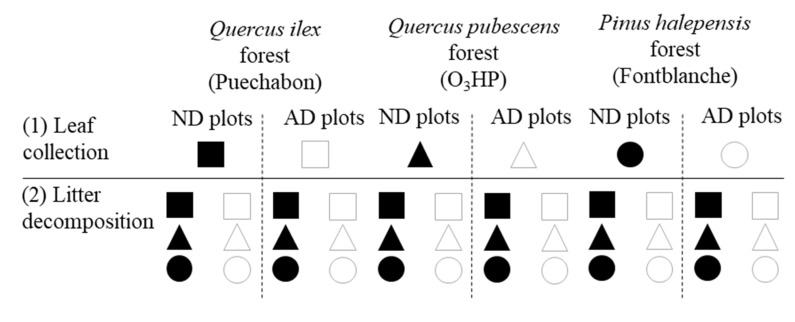
Litterbag experimental design. Three litter species (*Quercus ilex*, *Quercus pubescens*, and *Pinus halepensis*) coming from Natural Drought (ND) and Amplified Drought (AD) plots of their native forest (Puechabon, O_3_HP, and Font-blanche, respectively) were collected (1) and redistributed in the three forests and precipitation treatments (2) (according to [19]).

**Table 1 plants-11-02582-t001:** Initial quality of the three litter species according to the litter types. Values are mean ± SE (*n* = 5). *F*-values and associated-values are indicated. Different letters denote significant differences in initial litter quality with a > b > c. Significant differences in litter traits between litter types are in bold. *p*-values = ns: non-significant with *p* > 0.05, *: *p* < 0.05, **: *p* < 0.01, ***: *p* < 0.001.

Leave/Needles Quality	Litter Types (T)			Litter Species (S)								T × S	
		*F-Values*	*p-Values*	*Q. ilex*		*Q. pubescens*		*P. halepensis*		*F-Values*	*p-Values*	*F-Values*	*p-Values*
C (mg·g^−1^)	ND	0.01	ns	478.09 ± 1.35	b	462.61 ± 2.65	c	516.09 ± 1.60	a	172.61	***	0.16	ns
	AD			477.26 ± 1.29	b	462.11 ± 5.49	c	518.47 ± 2.90	a				
N (mg·g^−1^)	ND	8.67	**	**9.62 ± 0.20**	a	6.39 ± 0.20	b	5.36 ± 0.13	c	244.41	***	8.24	**
	AD			**8.22 ± 0.18**	a	6.17 ± 0.12	b	5.49 ± 0.09	c				
C/N (mg·g^−1^)	ND	9.97	**	**49.8 ± 1.00**	c	72.64 ± 2.31	b	96.41 ± 2.19	a	348.96	***	8.59	**
	AD			**58.17 ± 1.24**	c	74.92 ± 0.96	b	94.44 ± 1.39	a				
P (mg·g^−1^)	ND	**0.561**	**ns**	3.48 ± 0.21	a	1.90 ± 0.05	b	**1.56 ± 0.05**	c	198.85	***	7.58	**
	AD			3.23 ± 0.09	a	1.80 ± 0.08	b	**1.87 ± 0.02**	b				
Ca (mg·g^−1^)	ND	**80.56**	***	**25.09 ± 0.56**	b	**32.90 ± 0.77**	a	**18.41 ± 0.18**	c	487.43	***	0.104	ns
	AD			**21.68 ± 0.42**	b	**28.89 ± 0.58**	a	**16.10 ± 0.19**	c				
K (mg·g^−1^)	ND	12.12	**	**1.81 ± 0.06**	a	0.84 ± 0.01	b	0.95 ± 0.01	b	157.47	***	7.10	**
	AD			**2.77 ± 0.05**	a	0.91 ± 0.12	b	0.97 ± 0.02	b				
Mg (mg·g^−1^)	ND	82.74	***	**1.36 ± 0.01**	b	**2.83 ± 0.19**	a	1.48 ± 0.02	b	428.45	***	14.87	***
	AD			**1.87 ± 0.03**	b	**3.81 ± 0.08**	a	1.53 ± 0.01	c				
Na (mg·g^−1^)	ND	**27.6**	***	**0.11 ± 0.00**	a	**0.04 ± 0.00**	b	**0.11 ± 0.00**	a	249.11	***	55.05	***
	AD			**0.16 ± 0.01**	a	**0.07 ± 0.00**	b	**0.08 ± 0.00**	b				
Phenols (mg·g^−1^)	ND	34.88	***	**32.87 ± 1.58**	b	40.77 ± 2.59	a	38.59 ± 1.17	ab	2.63	ns	19.46	***
	AD			**58.48 ± 2.89**	a	44.18 ± 2.17	b	39.96 ± 1.49	b				
Lignin (mg·g^−1^)	ND	0.25	ns	337.38 ± 5.49	a	273.33 ± 9.00	b	302.57 ± 10.61	ab	22.11	***	0.992	ns
	AD			326.77 ± 6.24	a	283.64 ± 9.01	b	310.57 ± 5.66	ab				
Cellulose (mg·g^−1^)	ND	1.89	ns	204.99 ± 4.18	a	159.07 ± 8.01	b	150.81 ± 12.92	b	9.62	***	2.22	ns
	AD			198.16 ± 7.32		163.53 ± 5.93		182.75 ± 13.28					
Hemicellulose (mg·g^−1^)	ND	0.96	ns	270.02 ± 16.22	ab	274.22 ± 22.47	a	206.59 ± 10.42	b	18.40	***	0.812	ns
	AD			273.75 ± 11.97	a	309.58± 10.35	a	205.27 ± 12.22	b				
WSC (mg·g^−1^)	ND	**5.38**	*	187.61 ± 10.19	c	**293.38 ± 12.08**	b	**340.02 ± 7.13**	ab	74.53	***	4.89	*
	AD			201.32 ± 13.36	c	**243.24 ± 6.00**	b	**301.41 ± 6.07**	a				
WHC (%)	ND	18.12	***	**137.24 ± 0.98**	b	**146.90 ± 1.01**	a	113.89 ± 1.03	c	307.10	***	6.00	**
	AD			**129.89 ± 1.70**	b	**140.06 ± 0.75**	a	114.43 ± 1.38	c				
SLA (cm^2^·g^−1^)	ND	3.79	ns	**51.50 ± 0.88**	c	**133.13 ± 0.41**	a	83.50 ± 2.19	b	183.76	***	2.73	ns
	AD			**47.80 ± 0.87**	c	**128.16 ± 1.36**	a	85.06 ± 2.64	b				

**Table 2 plants-11-02582-t002:** PERMANOVA of the initial metabolomes of each litter species, according to litter types (Natural Drought (ND) and Amplified Drought (AD)). Degree of freedom (df), F-values, and associated *p*-values are indicated. *p*-values = ns: non-significant with *p* > 0.05, *.

		*Quercus ilex*		*Quercus pubescens*		*Pinus halepensis*	
	df	F-Values	*p*-Values	F-Values	*p*-Values	F-Values	*p*-Values
Litter types	1	9.23	0.013 *	2.44	0.034 *	0.97	0.464 ns

**Table 3 plants-11-02582-t003:** Examples of metabolites putatively identified from litter extracts of *Q. ilex* according to litter types (Natural Drought (ND) and Amplified Drought (AD)). RT: Retention Time in minutes, VIP: Variable Important in Projection. mSigma is a constructor quality value of the formula determination. The closer the value is to zero, the more accurate the result.

Litter Types	VIP	RT (Min)	*m/z* [M + H]^+^	*m/z* [M − H]^−^	Molecular Formula [M − H]^−^	Error (PPM)	mSigma	Putative Identification	Ref.
** *AD* **	M615T468	7.8	617.114	615.0987	C_28_H_23_O_16_	0.9	9.1	Quercetin-3-O-(6″-O-galloyl)-b-D-glucopyranoside	[44]
** *AD* **	M629T496	8.26	631.129	629.1144	C_29_H_25_O_16_	1.1	28	Isorhamnetin 3-(6″-galloylglucoside)	[44]
** *AD* **	M340T388 (isotope)	6.47	341.086	339.0717	C_15_H_15_O_9_	1.5	2.3	Aesculin	[45]
** *ND* **	M387T49	0.81	/	387.1143 [M + HCOO]^−^	C_13_H_23_O_13_	0.5	11	Disaccharide	
				341.1086	C_12_H_21_O_11_	0.9	13.8		
** *AD* **	M330T66 (isotope)	1.09	/	331.0671	C_13_H_15_O_10_	−0.5	21.6	galloyl-β-D-glucose	[44]
** *AD* **	M399T407	6.78	355.101	399.0934	C_16_H_17_O_9_	−3.9	NA	Chlorogenic acid	[46] [47] [48]
				353.0875	C_17_H_19_O_1_	−3.3	23.0		

**Table 4 plants-11-02582-t004:** Examples of metabolites putatively identified from litter extracts of *Q. pubescens* according to litter types (Natural Drought (ND) and Amplified Drought (AD)). RT: Retention Time in minutes, VIP: Variable Important in Projection. mSigma is a constructor quality value of the formula determination. The closer the value is to zero, the more accurate the result.

Litter Types	VIP	RT (min)	*m/z* [M + H]^+^	*m/z* [M − H]^−^	Molecular Formula [M − H]^−^	Error (PPM)	mSigma	Putative Identification	Ref.
** *AD* **	M711T533	8.88	/	711.3964 [M + HCOO]^−^	C_37_H_59_O_13_	−0.5	25.8	Arjungenin glycoside isomer 1	[49]
				665.3906	C_36_H_57_O_11_	4.6	na		
** *AD* **	M711T496	8.26	/	711.3965 [M + HCOO]^−^	C_37_H_59_0_13_	−0.9	14.9	Arjungenin glycoside isomer 2	[49]
				665.3906	C_36_H_57_O_11_	−5.5	425.9		
** *AD* **	M503T633	10.55	/	503.3382	C_30_H_47_O_6_	−0.5	3.9	Oleane triterpene	[50]
** *AD* **	M695T600	10.00	/	695.4018 [M + HCOO]^−^	C_37_H_59_O_12_	−1.4	6.9	Arjunglucoside II or arjunetin	[49] [51]
				649.3957	C_36_H_57_O_10_	−2.7	NA		
** *ND* **	M387T49	0.81	/	387.1144	C_13_H_23_O_13_	0.5	11	Disaccharide	[48]
** *ND* **	M702T383 (isotope)	6.38	/	701.5749 [M − H]^2−^	C_57_H_46_O_42_	0.9	220.4	Ellagitanin	[52]
** *AD* **	M593T520	8.66	/	593.1591 [M − H]^2−^	C_60_H_50_O_26_	−1.6	24.7	Proanthocyanidin (Cat-Cat-GalCat-GalCat)	[53]
** *AD* **	M343T637	10.61	/	343.0460	C_17_H_11_O_8_	0.3	4.8	Galloylquinic acid isomer	
** *AD* **	M289T405	6.74	291.086	289.0716	C_15_H_14_O_6_	1.6	11.4	(epi)catechin	[54]

**Table 5 plants-11-02582-t005:** Output of the most parsimonious model testing for the effects of litter species, litter types, precipitation treatments, forests, time of decomposition, and their interactions on litter mass remaining. D.f. = degrees of freedom, %SS = percentage of sums of squares. F-values and associated *p*-values are indicated (significant *p*-values are in bold). Full model: AIC = 3018.15, R^2^ = 0.60; Most parsimonious model: AIC = 2984.35, R^2^ = 0.59.

	d.f.	%SS	F-Value	*p*-Value
Litter species	2	1.9	9.5	**<0.0001**
Litter types	1	0.0	0.4	0.518
Precipitation treatments	1	1.6	16.4	**<0.0001**
Forests	2	4.2	21.6	**<0.0001**
Time	1	34.0	348.9	**<0.0001**
Litter species × Litter types	2	0.1	0.7	0.484
Litter species × Precipitation treatments	2	0.0	0.2	0.803
Litter species × Forests	4	2.7	6.8	**<0.0001**
Litter types × Forests	2	0.6	3	0.053
Precipitation Treatments × Forests	2	0.5	2.5	0.083
Litter species × Time	2	8.3	42.4	**<0.0001**
Precipitation treatments × Time	1	0.4	4.3	**0.039**
Forests × Time	2	3.8	19.6	**<0.0001**
Litter species × Litter types × Forests	4	0.8	2.2	0.073
Litter species × Precipitation treatments × Forest	4	1.0	2.6	**0.039**
Litter species × Forests × Time	4	1.0	2.6	**0.036**
Precipitation treatment × Forests × Time	2	0.6	3.2	**0.040**
Residuals	393	38.3	-	-

**Table 6 plants-11-02582-t006:** HFA according to the litter species and precipitation treatments after one and two years of decomposition. If ADH > 0, there is an effect of HFA; if ADH < 0, there is an effect of Home Field Disadvantage (HFD); if ADH = 0, there is no effect. *p*-values = **: *p* < 0.01, ***: *p* < 0.001.

Decomposition Time	Precipitation Treatments	ND			AD		
	Litter Species	*Q. ilex*	*Q. pubescens*	*P. halepensis*	*Q. ilex*	*Q. pubescens*	*P. halepensis*
First year	ADH	−3.55	−3.43	−4.92	−0.06	−4.35	−6.67
	T-value	1.24	2.41 **	1.84	0.03	3.36 ***	2.63 **
Second year	ADH	−0.30	−10.10	18.48	−13.15	−4.03	−1.59
	T-value	−0.04	−3.38 **	−3.46 **	0.14	−1.01	0.20

**Table 7 plants-11-02582-t007:** Climatic and topographic characteristics and composition of the three studied forests. Stand age: time elapsed since the end of management.

Forest	*Quercus ilex* L.	*Quercus pubescens Willd.*	*Pinus halepensis* *Mill.*
Site	Puechabon	Oak Observatory at the Observatoire de Haute Provence (O_3_HP)	Font-Blanche
Location	43°44′29″ N, 3035′45″ E	43°56′115″ N, 05°42′642″ E	43°14′27″ N 5°40′45″ E
Altitude a.s.l. (m)	270	650	425
Mediterranean bioclimatic zone	mesomediterranean	supramediterranean	thermos-mesomediterranean
Soil type	rhodo-chromic luvisol	pierric calcosol	leptosol
Soil texture	clay loam	clay	clay
Soil pH	6.60	6.76	6.80
Forest types	evergreen broadleaf	deciduous broadleaf	mixed coniferous/broadleaf
Other vegetation	*Buxus sempervirens* L., *Phyllirea latifolial* L., *Pistacia terebinthus* L. *Juniperus oxycedrus* L.	*Acer monspessulanum* L., *Cotinus coggygria Scop.*	*Quercus ilex* L., *Quercus coccifera* L. *And Phyllirea latifolia* L.
Tree density (stems/ha)	6070	5706	6000
Tree height (m)	5.5	5.5	13
Stand age (years)	74	70	61

## Data Availability

Metabolomics data will be archived in the Metabolights platform (https://www.ebi.ac.uk/metabolights/ accessed on 1 January 2017). Data concerning litter decomposition and litter quality will be archived in the SEDOO database from the BioDivMex Mistrals program (https://mistrals.sedoo.fr/BioDivMex/ accessed on 1 January 2017).

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
