# Peer review of "Amplified Drought Alters Leaf Litter Metabolome, Slows Down Litter Decomposition, and Modifies Home Field (Dis)Advantage in Three Mediterranean Forests"

_plants, 2022, doi:10.3390/plants11192582_

Round 1
Reviewer 1 Report
The research is relevant and interesting.
1. The titles of tables and figures are very long;
2. Method of Principal Component Analysis (PCA) need to be described more clearly;
3. I recommend not to use old literature sources;
4. Chapter "Material and Methods" must be moved after discussion.
Author Response
"Please see the attachment."

Reviewer 2 Report
This manuscript describes the effect of amplified drought on three forest species’ litter. I think this is a very well written manuscript with some English corrections that I have noted below. I feel that the descriptions and tables of the metabolites are too long and not necessary for this manuscript. It would be better to have a succinct summary of these metabolites and how they differ by treatment. I believe that this manuscript should be accepted after minor revisions.
Line 66: “indirectly affect”
Lines 74-76: Rephrase this sentence
Line 115: “Fixed upside down to not exclude”
Methods: It might be helpful to have a map of the three forests within France.
Line 163: Was it total phosphorus, Ca, K, Mg, and Na?
Line 167: “100”
Line 173: “compound”
Line 184: “weighed”
Tables 4 and 5: These tables might be better in an Appendix.
Lines 333-375: I feel that this information could be shortened or removed from the manuscript.
Line 418: “loses”
Line 432: “needle”
Line 436: “nutrient”
Line 442: “under increasing water stress”
Line 457: “antioxidant”
Line 492: “which may be due”
Lines 499-500: This study does not seem relevant here.
Line 550: “may negatively impact”
Author Response
"Please see the attachment."
